# Comparison of a Floating Cylinder with Solid and Water Ballast

**Roman Gabl** [1,2,*], **Thomas Davey** [1], **Edd Nixon** [1], **Jeffrey Steynor** [1] and **David M. Ingram** [1]

[1]   School of Engineering, Institute for Energy Systems, FloWave Ocean Energy Research Facility,
    The University of Edinburgh, Max Born Crescent, Edinburgh EH9 3BF, UK;
    tom.davey@flowave.ed.ac.uk (T.D.); Edward.Nixon@sintef.no (E.N.); Jeff.Steynor@ed.ac.uk (J.S.);
    david.ingram@ed.ac.uk (D.M.I.)
[2]   Unit of Hydraulic Engineering, University of Innsbruck, Technikerstraße 13, 6020 Innsbruck, Austria
[*]   Correspondence: roman.gabl@ed.ac.uk

**Abstract:** Modelling and understanding the motion of water filled floating objects is important for a wide range of applications including the behaviour of ships and floating platforms. Previous studies either investigated only small movements or applied a very specific (ship) geometry. The presented experiments are conducted using the simplified geometry of an open topped hollow cylinder ballasted to different displacements. Regular waves are used to excite the floating structure, which exhibits rotation angles of over 20 degrees and a heave motion double that of the wave amplitude. Four different drafts are investigated, each with two different ballast options: with (water) and without (solid) a free surface. The comparison shows a small difference in the body's three translational motions as well as the rotation around the normal axis to the water surface. Significant differences are observed in the rotation about the wave direction comparable to parametric rolling as seen in ships. The three bigger drafts with free surface switch the dominant global rotation direction from pitch to roll, which can clearly be attributed to the sloshing of the internal water. The presented study provides a new dataset and comparison of varying ballast types on device motions, which may be used for future validation experiments.

**Keywords:** floating cylinder; water filled; motion capturing; wave tank; wave gauges; fluid–structure interaction; free surface; sloshing

## 1. Introduction

The presented study investigates the response of a floating structure with two different ballast options, namely liquid (water) and solid. This allows the effects of the water sloshing to be isolated and quantified. Regular waves of constant amplitude and varying frequency $f_W$ are used to excite the motion of the simplified geometry. An additional background is presented in Gabl et al. [1] and the original data set is shared via the digital repository of the University of Edinburgh [2].

In general, water filled structures can be found in a wide range of applications including Wave Energy Converters (WEC) [3–5], Oscillating Water Columns (OWC) [6,7], and energy storage concepts [8,9]. The correct prediction of the motion and forces acting on a floating body are essential for the structural and mooring designs [10,11]. The response of Very Large Floating Structures (VLFS) [12–14], including the inner water level [15], have only been investigated for small motions. A wide range of studies investigate sloshing inside various containments and mainly focus on the resultant peak wall pressure [16–18], as well as being used for validation experiments [19–21]. The presented study aims to identify the influence of sloshing during large motions of a floating structure.

A comparable application of the investigated floating object can be found in ships filled with Liquefied Natural Gas (LNG) [22–26]. The influence of a rectangular tank on a rectangular barge is investigated by Su and Liu [27] based on a nonlinear Boussinesq-type, which failed in the case of steep sloshing phenomena. Huang et al. [28] also found significant limitations of the numerical approach in predicting large motions due to the influence of internal sloshing. Zhao and McPhail [29] compared frozen and liquid cargo in two spherical tanks on a vessel and identified a small difference in the rolling motion for half filled conditions. Bigger tanks are more susceptible to damage from pressure peaks caused by the internal sloshing [30,31]. Xu et al. [32] and Zhao et al. [33] investigated the side-by-side operation of two vessels through experiments at a scale of 1:60. A numerical simulation of such a combination is presented by Zhao et al. [34]. All these studies are focused on a very specific (ship) geometry and the presented work focuses on a more general approach by simplifying the floating structure to a hollow cylinder containing an internal water body and free surface.

Vortex-induced vibration on a cylinder is investigated for a wide range of geometries both experimentally and numerically [35–37] . Different commercial numerical products allow dynamic mesh zones in which the computational grid is reformed after every iteration. Zhu et al. [38] presented a recent study of an elliptic cylinder incorporating a two-way Fluid–Structure Interaction (FSI). The moving zone is in the range of three diameters for the comparable case of a cylinder mounted with fin-shaped strips [39]. Those studies, as well as the numerical sloshing experiment presented by Jamalabadi et al. [40], are only simulated as two-dimensional cases. Yang et al. [41] found a significant difference between the 2D- and 3D-approach for multi-bodies in a numerical wave tank. Different examples of a full coupling in 3D include propeller blades [42], offshore wind turbine towers [43], OWC [44], and point-absorbing WEC [45]. Alternatively, FLOW-3D leaves the grid constant and changes the discretisation of the geometry depending on the object's movement [46–48].

Meshless methods, such as Smoothed Particle Hydrodynamics (SPH), which are based on a purely Lagrangian technique, are a possible alternative option for large motions. Successful applications can be found from the simulation of the flow around cylinder incorporating an erosion model [49], the design of a floating tsunami shelter [50], the simulation of complex wave tanks [51], and the coupling with wave propagation models for wide areas [52]. Different studies apply this method for the simulation of a floating buoy [53] and WEC [54]. Some studies investigate damaged ships [55], which includes one large chamber with a hole [56] as well as simulations of one of three separate ones [57,58]. The comparison with experimental observations show good agreement for these cases, which makes SPH a valuable option to simulate the presented big motions of a floating structure.

This paper presents the experimental results of a simplified geometry floating in a wave tank under regular wave conditions. Consequently, the response of the floating object is influenced by the inner and the outer water bodies. This adds a further complexity to those sloshing experiments caused by excitation of the containment [19–21]. The aim of this study is to identify the influence of the internal liquid by comparing it to a solid ballast option. Four different drafts are investigated and the response in all six degrees of freedom are separately compared and presented in the following sections. The complete data-set is available according to Gabl et al. [1,2], and the results found are used to refine future measurement with the water filled cylinder. The presentation of this data set and experimental results will greatly benefit the numerical modelling community, who require a validation case for new solutions to capture this phenomenon.

## 2. Experimental Set-Up

### 2.1. General

The investigated geometry is a floating cylinder with an outer diameter $D_{out}$ of 0.5 m and a total height $H$ of 0.5 m. Figure 1 shows the transparent structure, which has wall thicknesses of 5 mm and a 7 mm floor plate. Wave gauges (WG) made of copper tape are applied on the inside wall. This instrumentation is introduced by Gabl et al. [59] and allows the measurement of internal run up at the

walls. In the presented investigation, the movement of the internal water surface is not specifically monitored, hence the main goal is the comparison of the body motions between the ballast variations, namely water or solid, for a wide range of wave conditions. This allows the identification of the interesting cases where future additional investigations of the water ballast may be performed.

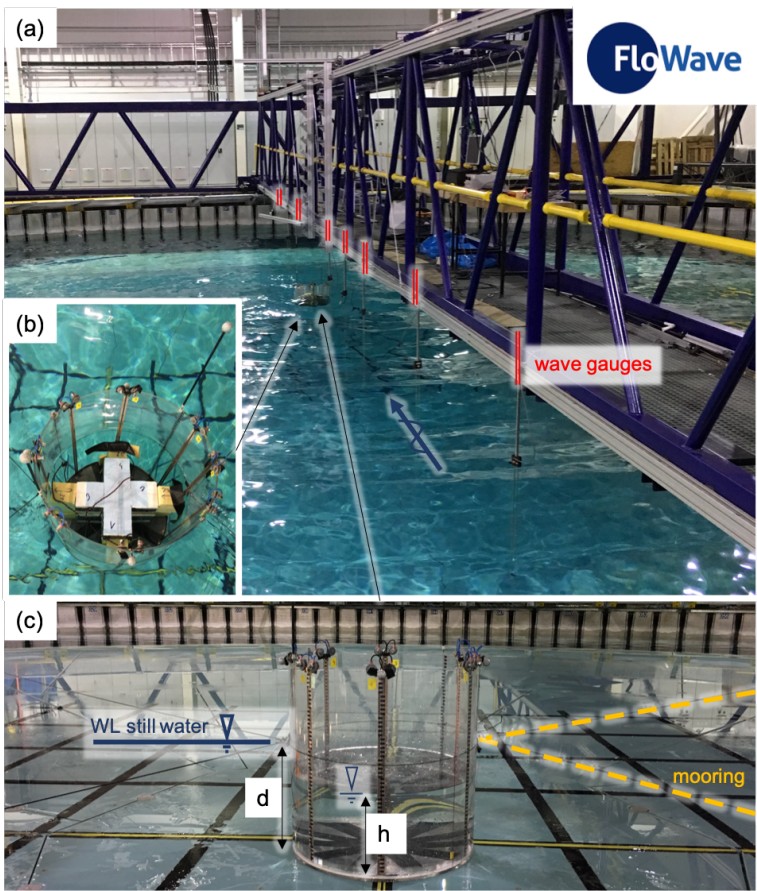

**Figure 1.** (**a**) overview of experimental set-up in the wave tank including wave gauges and main wave direction; (**b**) detail of the solid ballast; (**c**) cylinder filled with inner water on the raised tank floor with connected mooring lines.

Six markers are mounted at the top of the structure to use an optical motion capture system (Section 2.2). This in turn allows the 6 degree-of-freedom measurement of the motions of the floating cylinder under different wave conditions. The mass of all additional loads (measurement equipment) on the structure are reduced as far as possible but need to be located at the dry top. Consequently, the initial structure was observed to be unstable at smaller displacements (or low inner water depth $h$ values). A reduction in the body's vertical centre of gravity was therefore required to allow a wider range of $h$ values. A layer of lead ballast with a thickness of approximately 2.6 mm was located at the bottom inside of the cylinder. This had an added advantage in that it concealed a drain plug in the centre of the floor plate ($R_{Plug}$ = 5 cm) and thus provided an uninterrupted plane on the internal floor.

The floating cylinder was tested in the FloWave test basin at the University of Edinburgh, which allows the combination current with non-directional waves [60–63]. The model was moored in the centre of the circular wave tank which has a diameter of 25 m and a depth of 2 m (upper volume of the tank). For the presented investigations, long crested sinusoidal wave inputs were applied with an initial wave amplitude of 0.05 m and a frequency range $f_W$ covering 0.3 Hz to 1.1 Hz parallel to the gantry (Figure 1a). A set of 16 runs were predefined to cover the whole frequency band. Depending on a preliminary analysis, further cases were individually added to investigate specific observed behaviour and refine near resonances. The test length was 180 sec in total starting simultaneously with

the wave makers. Between the tests, a settling time was allowed so that the motions of the floating cylinder were negligibly small ($\leq$0.5 mm).

The presented investigation focuses on the comparison of the solid ballast case for a range of water volumes inside of the cylinder. The solid ballast modules are located so as to maintain symmetry in the two main axes parallel to the still water surface (Figure 1b). A layer of Styropor (an expandable polystyrene material) was added on the top and bottom of the resulting cross arrangement so that the vertical centre of gravity is the same as that of the still water ballast case. In total, three different masses are combined (two crosses and one cylinder, the exact masses can be found in Gabl et al. [1]), which allowed comparison of a wide range of different inner water levels. The water filled case is investigated after the solid case and the exact displacement and external still water line is reproduced. Figure 1c shows the model on the raised tank floor with an indicative inner water level h.

Table 1 lists the four different cases, which are defined by the draft (vertical distance between the bottom of the cylinder and the external still water level). The distinction between the four draft options is made based on the comparison to the minimal draft $d = d_{Min}$ of 0.1822 m. The inner water height $h$ is defined as the vertical distance between the internal bottom of the cylinder and the internal still water level. This value changes from nearly a quarter to more than a half of the total height of the cylinder $H$. The observed motions are comparably large but an exchange of water between inner and outer water body—spilling and over topping—did not occur.

**Table 1.** List of the four different investigated cases with the corresponding water depth $h$ in relation to the total height $H$ of the cylinder and the resulting draft $d$.

|  | $d \times 1.0 = d_{Min}$ | $d \cdot 1.25$ | $d \cdot 1.5$ | $d \cdot 1.75$ |
|---|---|---|---|---|
| Total Mass (kg) | 35.70 | 44.65 | 53.50 | 62.45 |
| h (m) | 0.1232 | 0.1707 | 0.2178 | 0.2653 |
| h/H (–) | 0.25 | 0.34 | 0.44 | 0.53 |
| draft (m) | 0.1822 | 0.2279 | 0.2730 | 0.3187 |

*2.2. Measurement*

Two different measurement systems are used: (a) wave gauges and (b) an optical motion capture system. In total, seven wave gauges are aligned parallel to the direction of wave propagation through the centre of the tank. This allows the surface elevation in the tank to be measured at discrete points, five in front and two behind the floating cylinder. At the beginning of each test day, all probes are calibrated to maintain the high accuracy of the measurements. The accuracy of a resistive wave gauge is typically in the range of 1 mm [59,64,65].

The motion of the floating cylinder is measured with the motion capture system installed in the FloWave facility. This includes eight Qualisys cameras (Göteborg, Sweden) , which are distributed over one side of the test tank at different heights. The global coordinate system is orientated as part of the calibration process of the motion capturing system in the centre of the round wave tank. The positive x-axis is defined parallel to the gantry, which is identical to the wave direction used in these tests. Between each tested experimental set-up, the calibration is refined so that the overall accuracy of the localisation of the points could be kept smaller than 1 mm.

The 3D-locations of each point are assembled together to calculate the motion of the floating cylinder as part of the post-processing of the motion capturing software, Qualisys Track Manager (QTM, version 2019.3, Qualisys, Göteborg, Sweden) . Therefore, a body definition is needed, which connects the local body coordinate system to the measured markers and is dependent on each investigated experimental set-up (draft). Therewith, it is assured that the vertical position of the origin of the local body coordinate system is always level with the still water surface in the wave tank. All local axes are orientated similarly to the global coordinate system. The rotational symmetry of the cylinder is a big advantage over more specific ship geometries by providing independence to orientation of the incoming waves. Furthermore, a small rotation around the z-axis has nearly no influence on the

behaviour of the floating structure in relation to the wave. The post-processing software, QTM, also allows for identify pitch and roll angles independently of yaw, thereby simplifying the analysis in this case. Consequently, the measured rotations for pitch and roll are always co-linear with the global tank coordinates, and therefore fixed relative to the wave direction, regardless of the cylinder rotation.

All instrumentation is triggered by a Transistor—transistor logic (TTL) pulse upon the start of wave generation and operate at the same measurement frequency of 128 Hz. Further information is given in Gabl et al. [1], which describes the available full data set of this investigation.

### 2.3. Mooring

A station keeping mooring system is necessary, but it should be soft enough not to introduce any effects on the wave frequency motion response. A horizontal approach near the water surface was chosen, which is comparable to Zhao et al. [66], who investigated the roll motion of a barged vessel with two cylindrical tanks. Four lines are connected to the quadrants of the round tank in 45° to the main axes (Figure 1c). Each one consists of a hollow elastic of 3 m long (diameter 3 mm) with a very high stretch factor and is expanded with standard rope (6.5 m) to the tank side. A small but balanced preload was introduced. Two of the mooring lines are joined together in one point with a symmetry along the $x$–$z$-plane. Both connection points are adjustable with the draft (Figure 1c) so that the rotation axis introduced by the mooring system is equal to the local $y$-axis (in the height of the still water surface). This minimises the influence for the main rotational motions (pitch; around the $y$-axis). The roll motion is affected more significantly, but the mooring set-up is generally very soft. The natural frequency of the mooring system is presented in Section 3.2.

## 3. Results

### 3.1. Overview

The analysis of the results is split into four parts. First, the different experimental set-ups are investigated in the still wave tank. In this case, the floating structure is manually excited in a series of free oscillation tests to verify the model's behaviour in connection with the mooring system (Section 3.2). All further analysis discusses the results of the response motion of the different drafts and ballast options. As an overview of all six degrees of freedom, Section 3.3 presents the minimum and maximum values after the initial start-up period. The following detailed analysis of the translation in the $z$-direction is shown in Section 3.4. Furthermore, it was observed that significant roll motion is connected to a reduction in pitch. Based on this, both rotations are presented together in Section 3.5. Section 3.6 summarises the results with the different drafts and allows a direct comparison between the liquid and solid ballasts.

### 3.2. Free Oscillation

For these tests, the cylinder is manually excited out of its static equilibrium state in the centre of a still water tank. The oscillation is captured for each degree of freedom and repeated multiple times. Figure 2 presents the response frequency $f$ for the two different ballast configurations (solid and liquid) in relation to the draft $d$ (normalised by the minimum draft $d_{Min}$ of 18.22 cm).

In surge and sway, the response is dominated by the mooring lines and is very slow. Consequently the frequency is small and decreases as draft increases. A comparable behaviour—but at a higher frequency level—can be observed in the $z$-direction (heave) with a response of around 0.9 Hz for the minimal draft and 0.7 Hz for the heaviest ballast. For all translational motion, the difference between the two ballast configurations is not significant.

The tilting rotation around the $x$-axis is slightly more restricted than around the $y$-one, a deliberate feature of mooring setup intended to result in the smallest possible influence in pitch (Section 2.3). As expected, the water filled version of the floating cylinder shows a different response in roll and pitch. For the solid case, the changes due to a variation in the draft are relatively small and a typical

value is around 0.9 Hz. In contrast to this, the sloshing inner water leads to a significantly smaller frequency and also a bigger dependence. The frequency is nearly doubled for the bigger draft in comparison to the version with the minimal draft $d_{min}$ of 18.22 cm. A small difference can be observed for the rotation around the $z$-axis with the draft having nearly no influence.

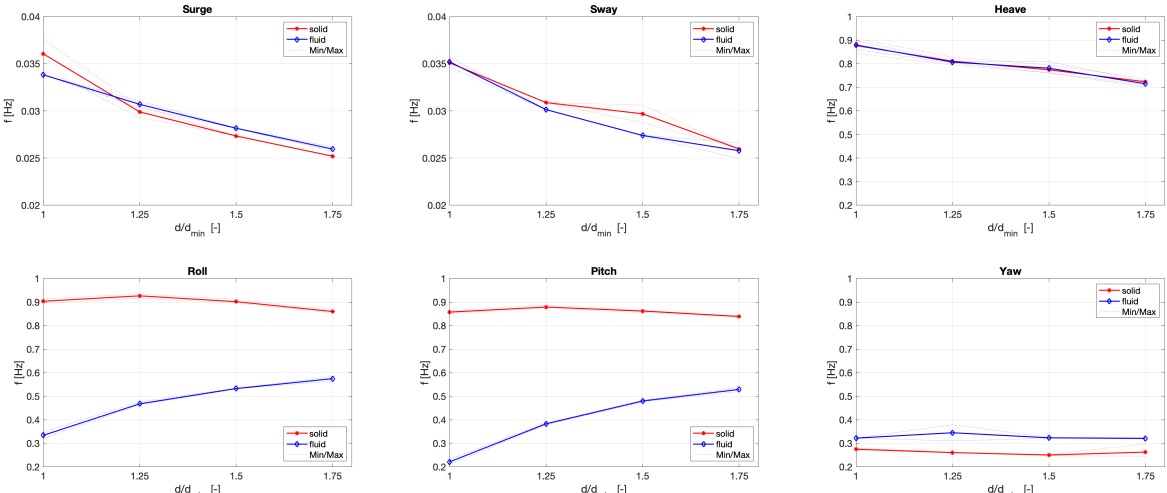

**Figure 2.** Response frequency $f$ of the free oscillation experiments for each degree of freedom (upper row movements, lower row rotations)—comparison of solid and liquid ballast for four draft cases $d$ normalised by $d_{min}$ = 18.22 cm.

### 3.3. Minimum and Maximum Response Motion

The response motion of the floating cylinder under wave conditions is analysed for all six degrees of freedom separately. Figure 3 shows the results for the movement and the complementary rotations are presented in Figure 4. In all cases, the first 52 s from the starting of the wave makers are excluded to limit the analysis to the fully developed oscillation of the floating cylinder in the wave tank. This has the advantage that delayed effects caused by the sloshing of the inner water body are fully developed. On the other hand, due to the different final position of the cylinder in the tank (especially the $x$-direction, Figure 3), the reflections can have a different influence on the response of the floating structure [60]. The chosen approach is focused on an overall comparison. For a specific validation in the time-domain, the original measurement [1,2] should be checked for transient behaviour (especially switching between roll and pitch), and it is advisable to focus on the ramp up as well as the first stable oscillations [67].

In the following figures, the dashed line shows the minimum and uses the same colour as the maximum line. The upper row of graphics present the solid case and the lower one the ballast option with water. Generally, it can be noticed that the response is relatively small up to 0.6 Hz and peaks at around a range of 0.7 Hz to 0.9 Hz.

The biggest absolute movements can be found in $x$-direction (surge; wave direction), which is independent of the investigated option (Figure 3). The floating model drifts until the restricting forces of the mooring lines provide a new equilibrium. This drift reaches up to 2 m in surge, but the individual oscillations range up to around 200 mm. This behaviour is as expected due to the very soft mooring system. The model oscillates also in the approximate range of 200 mm in the orthogonal direction parallel to the $y$-axis. In heave, the movement can reach up to the double the wave amplitude of 50 mm (the presented results are limited to this main value). The observed response in the $z$-direction is analysed in detail in Section 3.4.

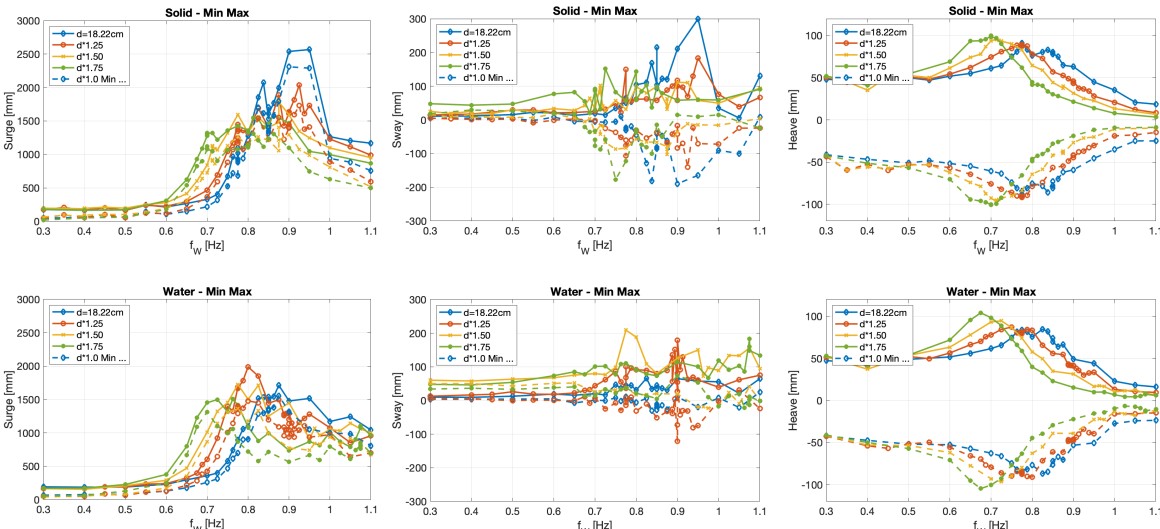

**Figure 3.** Summary of the minimum and maximum value of the movement in *x*-(Surge), *y* (Sway) and *z*-direction (heave)—values in mm—Solid (upper) and liquid (lower) case in relation to the initial wave frequency $f_W$.

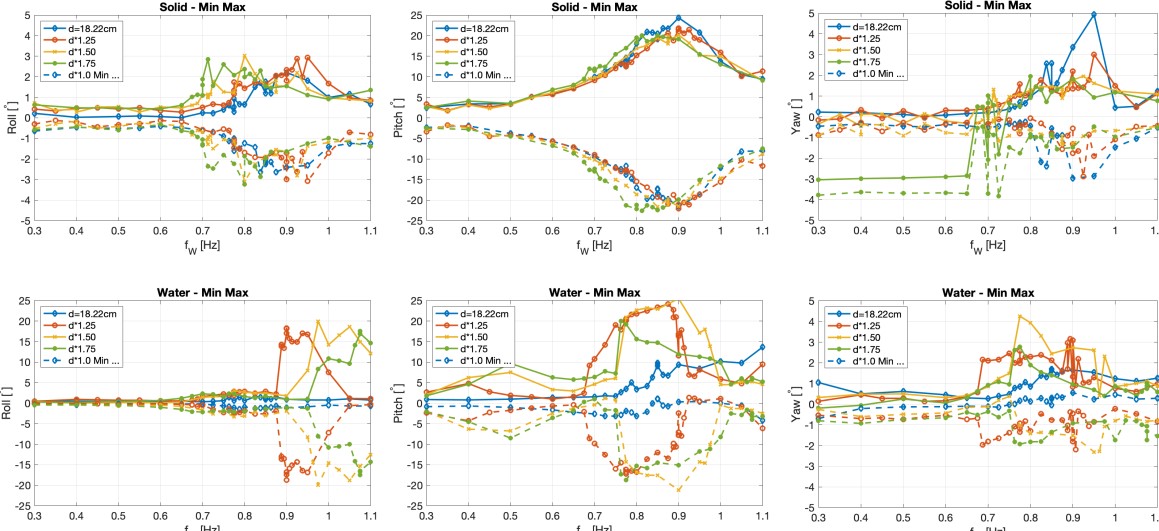

**Figure 4.** Summary of the minimum and maximum value of the rotation around the *x*- (Roll), *y*- (Pitch) and *z*-axis (Yaw)—values in °—Solid (upper) and fluid (lower) case in relation to the initial wave frequency $f_W$.

The solid ballast cases, in addition to the smallest water draft, have similarly small rotations around the *x*-axis (Roll). The range is approximately 3° to 4°. In contrast to this, the three cases with more water inside the cylinder have a very clear response in roll, which will be discussed in Section 3.5. The response in pitch peaks around a wave frequency $f_W$ of 0.8 to 0.9 Hz. With up to 25°, the rotations are large, but nevertheless no spilling out or overtopping could be observed.

As mentioned in Section 2.2, the post-processing resolves the two main rotations around the *x*- and *y*-axis independently of a rotation around the vertical axis. Figure 4 shows that the angles are nevertheless very small as it is expected for a rotational symmetric geometry. For the solid case with the draft of $d \cdot 1.75$, the model was turned approximately 3° for the first tests with a lower wave frequency $f_W$, which was corrected (Figure 4; right graph in the upper row—Yaw Solid—green lines). A repetition was not conducted; hence, the analyses are not sensitive to this rotation.

For all three translational degrees of freedom (Figure 3), the difference between the two investigated ballasts are comparably small. A significantly different response could be identified for the three deeper drafts in the roll motion (Figure 4), which could not be observed in the solid case. Further analyses are limited to the vertical movement (Section 3.4) of the floating cylinder as well as the two main rotations (Section 3.5).

*3.4. Heave Response*

Based on the overview of the three different motions in Figure 3, the movement in the vertical direction is chosen to be presented in detail. For all following analysis, it is assumed that the response of the floating structure can be described as a (sum of) sine oscillation(s) similar to the incoming sine wave. The responding amplitude as well as the frequency $f_R$ is evaluated based on the initial wave frequency $f_W$ produced by the wave makers.

Similar to Section 3.3, the first 52 s of the total measured data (180 s) is excluded to ensure the system has reached steady state. The remaining period is split in eight equal sub-datasets for the following evaluation. A sine function is fitted to each subset separately. The analysis code (developed in MATLAB (version R2019a, MathWorks, Natick, Massachusetts, USA) considers a least-squares cost function as part of the fitting process. Incorrect results for the amplitudes are automatically sorted out based on the relation to the range of the local minimum/maximum of the measured data as well as the Root Mean Square Error (RMSE). The method was repeated for 4, 16 and 32 interval cases, which showed either a worse fitting rate for specific frequencies or only insignificant differences to the chosen approach with 8. The gained amplitudes of all sub-data sets are average for each initial wave frequency $f_W$. The results are verified with a single Fast Fourier transform (FFT) analysis [67], which showed a very good agreement.

In case of heave, those values for the amplitude are normalised by the measured wave amplitude $a_W$ in front of the floating cylinder (details are given in Gabl et al. [1]) and therewith presented in Figure 5. The upper row shows the results for the solid ballast option and the lower row the complimentary ones for the cylinder filled with water. It is evident that all investigated cases indicate a comparable behaviour for the heave response. In the lower frequency band, the floating cylinder moves equal to the incoming waves and the amplitudes exceed those value with a peak in the range of 0.7 to 0.9 Hz. In this case the response frequency observed in the free oscillation experiments (Section 3.2, Figure 2) is reproduced, which leads to resonance. The observed movements reach up to double the wave amplitude. Higher frequencies lead to a near decoupling of the response and the movement of the floating cylinder decreases significantly. Consequently, the investigated wave frequency covers the full spectrum.

The response of the floating cylinder shows in some frequencies significant additional oscillations. This can also be observed based on the Fast Fourier transformation of the measurement. To isolate those additional oscillations, the fitted sine function (first step presented as a blue line) is subtracted from the original data, and a further fitting is conducted for the modified dataset. This process is repeated until no significant fitting is possible. The results are presented as 2nd and 3rd peaks in the following graphics. In the lower frequencies as well as in the range around the response frequency of the free oscillation (Section 3.2), significant secondary phenomena are detectable.

Figure 6 presents the response frequency $f_R$ and includes indicative lines for the wave frequency multiplied with different constants. For the first sine fitting, the frequency $f_R^{1st}$ is identical to the incoming $f_W$. Only in the highest frequencies is a small deviation found. This is caused by reflection in the wave tank, hence those high frequency waves cannot be absorbed by the receiving wave-maker paddles. For the solid ballast option (Figure 6, upper row), the second successful sine fitting has a doubled response frequency relative to the first one (equal to $f_W$). This is also true for the lower frequencies for the water filled ballast option. A factor of 0.5 and 1.5 could be identified, which is assumed to be caused by the sloshing of the inner water stored in the cylindrical tank.

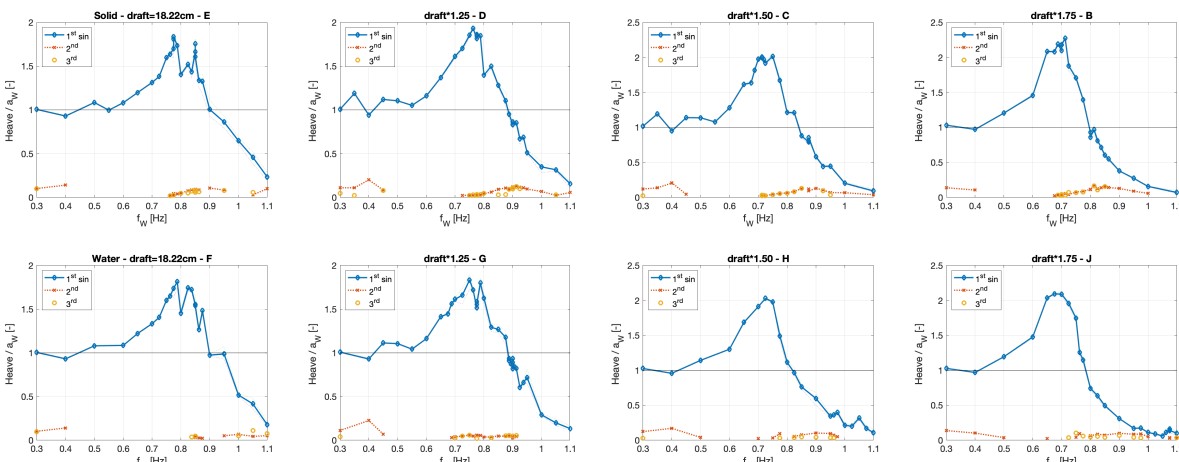

**Figure 5.** Amplitudes of the sine fitting for the movement in the $z$-direction (heave) normalised by the measured wave amplitude $a_W$ in relation to the wave frequency $f_W$—upper row solid ballast option of the four different drafts and the corresponding water filling in the lower row.

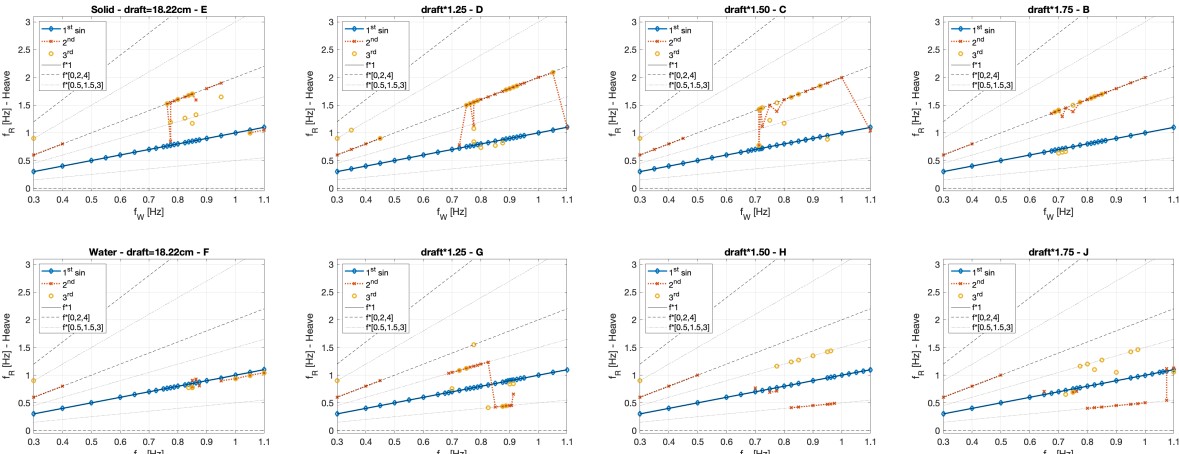

**Figure 6.** Response frequency $f_R$ of the sine fitting for the movement in the $z$-direction (heave) in relation to the wave frequency $f_W$—upper row solid ballast option of the four different drafts and the corresponding water filling in the lower row.

Generally, the differences between the to ballast options, namely the solid case and the cylinder filled with water, are very small for the movements in the $z$-direction (heave) and more significant differences can be found in the rotations presented in Section 3.5.

### 3.5. Roll and Pitch Response

As shown in Section 3.3, the rotation around the $z$-axis is very small and is not significant for the analysis of the other degrees of freedom based on the rotationally symmetrical design of the floating cylinder. Figure 7 presents the results of the different stages of the sine fitting for the rotation around the $x$-axis (roll) and those around the $y$-axis (pitch) in Figure 8. The evaluation process is similar to the analysis of heave, which is described in Section 3.4. In these particular graphs, the measured unit [°] is chosen and frequency bands with a response smaller than 0.5° are marked grey.

Two specific characteristics can be observed for the pitch rotation in Figure 8. The water filled cylinder has a first response peak in the range of the natural frequency (Section 3.2, Figure 2), which has to be investigated in further tests. A second one is in the range of the double response frequency $f_R$ based on the free oscillation test, which is in the same range as the $f_R$ of the solid ballast option. This analysis is supported by Figure 10. The first sine fitting for the solid case shows a direct connection

between the wave frequency and the response, but, for the water filled variation, the primary oscillation is connected with half of the $f_W$.

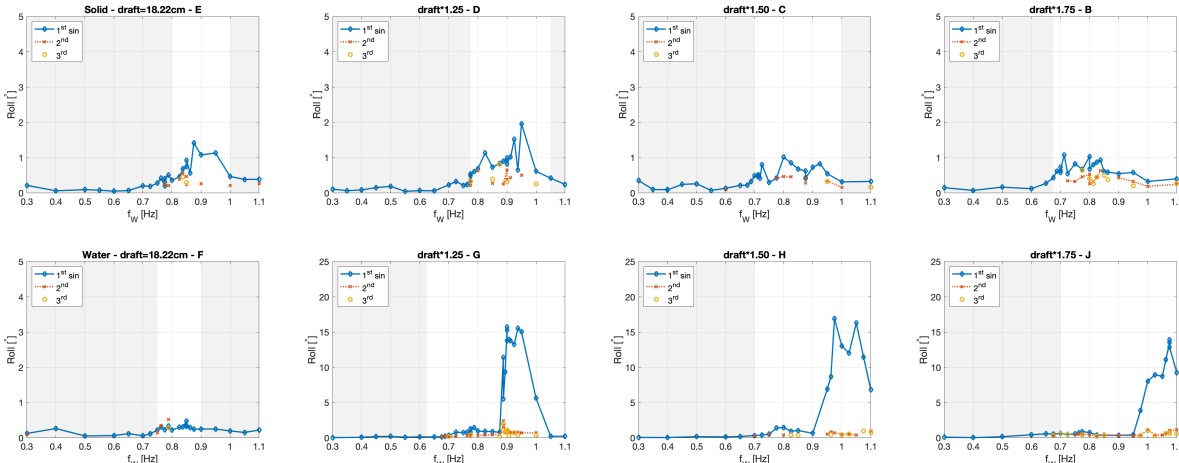

**Figure 7.** Amplitudes of the sine fitting for the rotation around *x*-direction (roll) in relation to the wave frequency $f_W$—upper row solid ballast option of the four different drafts and the corresponding water ballast in the lower row.

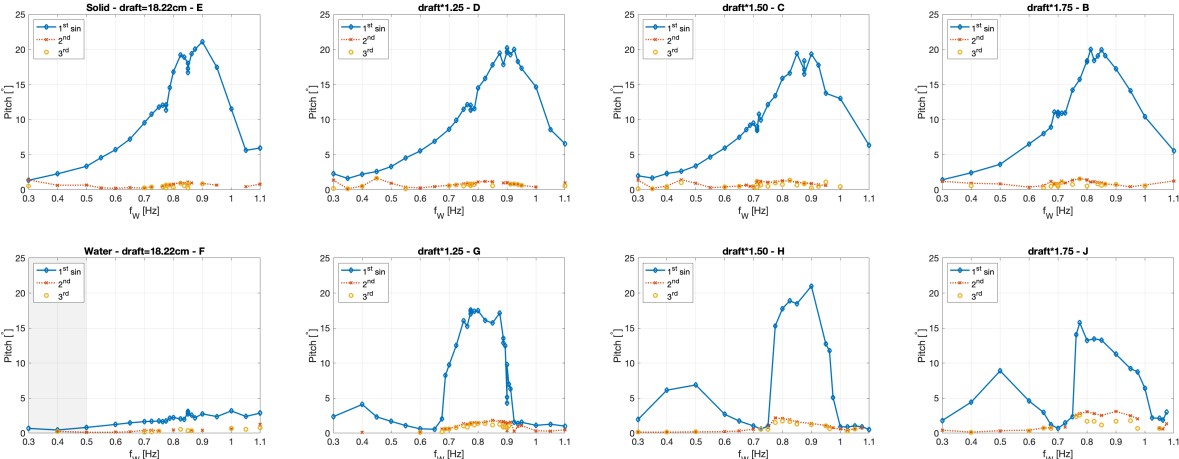

**Figure 8.** Amplitudes of the sine fitting for the rotation around *y*-direction (pitch) in relation to the wave frequency $f_W$—upper row solid ballast option of the four different drafts and the corresponding water filling in the lower row.

In most of the cases, the roll motion is comparably small (Figure 7), except for the three deepest draft water ballast cases. In these cases, the pitch motion of the water filled cylinder decreases significantly as the roll motion develops. Roll angles over 20° are observed. In the transient frequency zone, multiple switches between the two rotations direction can be observed. This indicates that the response caused by the incoming waves is not stable, clearly caused by the sloshing inside of the floating cylinder. This leads to an unstable phenomenon, which can be compared with parametric rolling of ships. Such a stability variation occurs based on the governing parameter of wave height and length as well as the ship geometry [68–70]. It can be reduced by changes of speed and direction to limit the loads in the securing system of container ships [71,72].

The analysis of the corresponding response frequency for roll (Figure 9) and pitch (Figure 10) indicates that the cylinder oscillates in both directions at half of the wave frequency $f_W$. France et al. [68] show that the parametric rolling in ships occurs in the ratio 2:1 of the period of pitch and roll, which is not the case for the tested floating cylinder. Consequently, the observed phenomenon is only similar to

parametric rolling and more likely connected to a resonance to the natural response frequency found in the free oscillation (Section 3.2, Figure 2). Further investigations are needed to study this in detail.

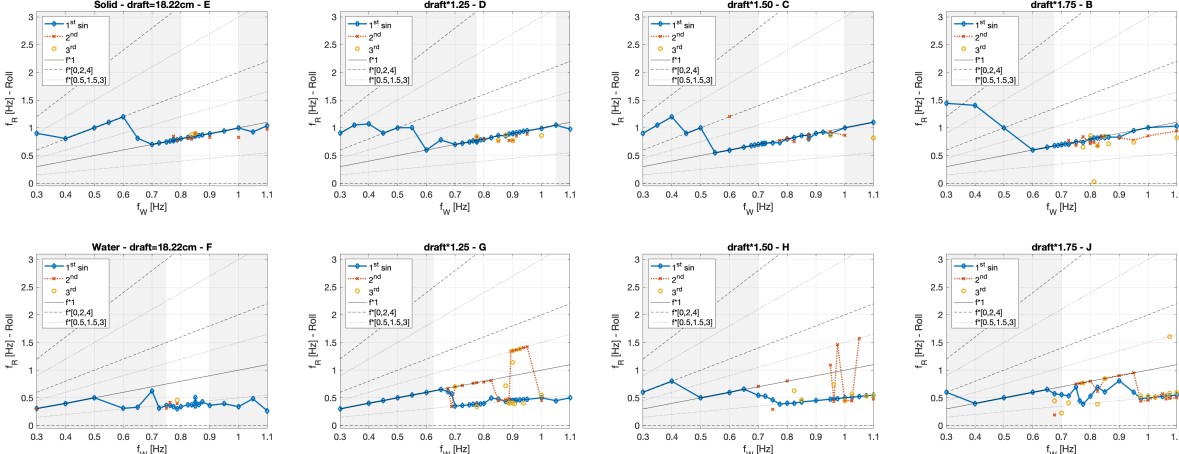

**Figure 9.** Response frequency $f_R$ of the sine fitting for the rotation around the *x*-direction (roll) in relation to wave frequency $f_W$—upper row solid ballast option of the four different drafts and the corresponding water filling in the lower row.

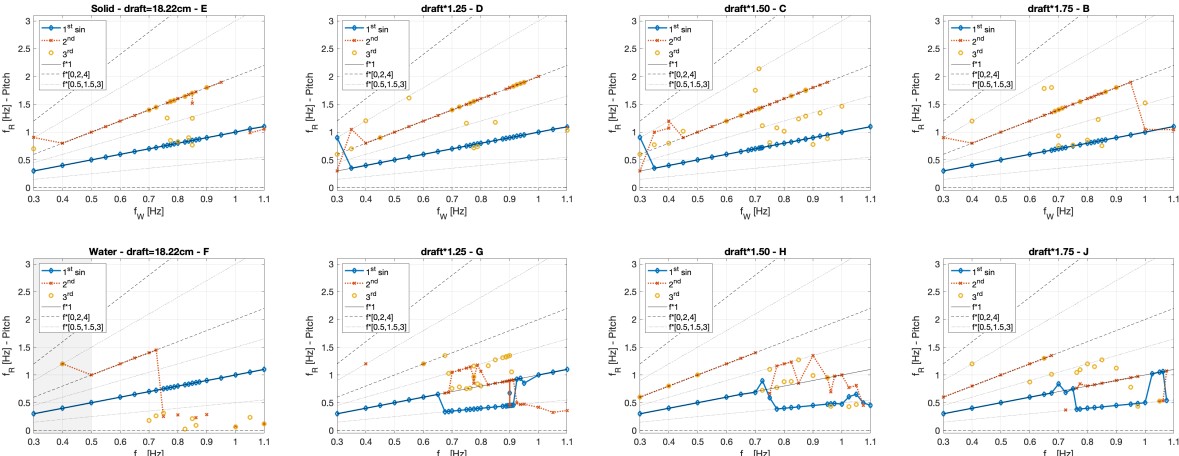

**Figure 10.** Response frequency $f_R$ of the sine fitting for the rotation around the *y*-direction (pitch) in relation to wave frequency $f_W$—upper row solid ballast option of the four different drafts and the corresponding water filling in the lower row.

For the experimental set-up filled with the deepest inner water level *h*, and hence the biggest draft, this response is observed in a $f_W$-range near the absorption capacity of the wave tank. Consequently, the response is not as clear as it would be without this disturbance. Nevertheless, it is important to mention that this effect also occurs for all three deeper draft cases, but, for further investigations, smaller inside water levels are better suited for investigations in FloWave. In contrast to the small roll angles, the pitch results are significantly larger. Only the lowest water ballast draft configuration reaches a nearly stable condition in the incoming waves and shows nearly no motion. For all other investigated cases, the goal to investigate big motions are reached.

### 3.6. Summary and Comparison of the Two Different Ballast Options

In Sections 3.4 and 3.5, each of the four investigated drafts is presented separately. Figure 11 summarises the amplitudes of the first sine fitting for heave (Figure 5), roll (Figure 7) and pitch

(Figure 8). The splitting between the two ballast options, solid (upper row) and water, is maintained as well as the line colour of the Figures in Section 3.3.

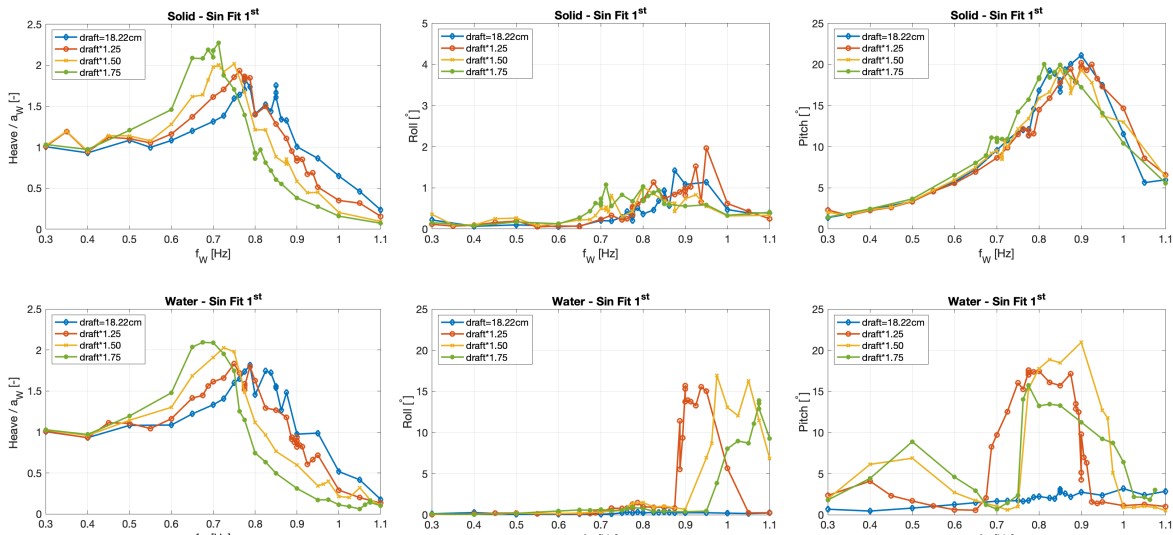

**Figure 11.** Summary of the fitting with sinusoidal functions for the movement in $z$-direction (heave normalised by the measured wave amplitude $a_W$) and rotation around $x$-(Roll) and $y$-axis (Pitch)—Solid (upper) and fluid (lower) case in relation to the initial wave frequency $f_W$.

The responding movement of the floating cylinder in the $z$-direction (heave) is for both ballast options directly connected to the wave frequency $f_W$ of the tank. The peak values of the normalised response is in the range of more than double the incoming wave amplitude and increases with a bigger draft (distance between the outside bottom of the floating cylinder to the still water surface in the wave tank). The shift of the peak frequency is similar to the findings of the free oscillation experiments (a bigger draft results in smaller response frequency; Section 3.2 and Figure 2), which also shows no significant difference between the two ballast options. Comparable results are found by Xu et al. [32] for the side-by-side arrangement of a floating LNG tanker.

A different response to the incoming waves can be observed in the roll motion (rotation around the $x$-axis). Nearly no rolling motion occurs for the solid cases as well as the smallest draft for the liquid ballast configuration. In contrast to this, the three set-ups filled with water oscillate orthogonal to the incoming wave front instead of in the wave direction. This behaviour indicates the occurring of an instability phenomenon comparable to parametric rolling in ships [68]. The shift in the peaks of this roll motion also correspond with the change of the response frequency of the free oscillation (a deeper draft leads to a bigger frequency), but occurs at a wave frequency, which is approximately the doubled response frequency. In a transient phase, both rotations can be alternately observed. This effect is clearly initiated by the sloshing inner water body, which is stored inside of the floating cylinder. A further investigation of this phenomena has to be conducted, which includes a detailed monitoring of the inner water surface.

The set-up with the shallowest inner water level is the only one that doesn't respond with a significant pitch angle. For all other cases, values of over $20°$ are observed and therewith the requested big motions reached. Nearly no difference between the different draft variations are found for the solid ballast option. The three corresponding water filled set-ups reach nearly the same maximum level, but, in the higher frequency band, the rotation starts to switch to orthogonal direction. These three water filled cylinders also respond in an additional peak when the wave frequency is equal to the response frequency found in the free oscillation. The deeper draft corresponds with a higher frequency. This frequency band should be further investigated to clarify this effect.

## 4. Discussion

The main goal of this investigation is to gain an overview of the behaviour of water filled floating cylinders under wave conditions and large motions. Four different solid cases with a similar mass distribution are investigated and compared to the water filled variation.

A cylinder is chosen as a simplified geometry for this investigation. Keeping typical space discretisations in mind, this is not ideal for further comparison with numerical simulations. On the other hand, this approach reduces the influence of the orientation of the model to the incoming waves to a minimum as well as vortexes caused by sharp edges. From a hydrodynamic point of view, an addition of a hemisphere on the bottom of the cylinder would be a good addition, thereby avoiding the sharp edge between the side wall and the bottom surface. This option can be investigated in a further measurement campaign along with different cylinder diameters.

In nearly all cases, a stable oscillation could be observed for the response of the floating object. The single exception is the transient phase for the water filled variation, when the motion switches between pitch and roll. The rotational symmetry also allows the use of the post-processing features of the Qualisys system to evaluate pitch and roll motion independently of a rotation around the vertical $z$-axis. Nevertheless, the yaw angles are small.

An important point for a good comparison is the solid representation of the water. This approach should have the mass distribution but no influence caused by sloshing. The first approach considered was to fill the cylinder with a gravel mixture of an equal density. This would have the big advantage that a wider range of inner water levels could be investigated. On the downside, it is also possible that such a loose material could start to move inside the cylinder (even if this would be significantly smaller than the liquid case). An accidental pollution of the water in the wave tank is also possible, which has the potential for major disruption to FloWave's complex wave and current generation systems. Furthermore, it was considered that a mixture of the correct density would be hard to reproduce and guarantee that it stays homogeneous. Therefore, the option of building fixed ballast options based on iron blocks is chosen. The two crosses only allow a symmetrical distribution according the two main axes, but can be exchanged very quickly and repeatably. In addition, a cylindrical weight is used to combine the four different draft variations. The investigation showed no significant influence based on this assumption.

The positioning of the marker point on the top of the cylinder is not ideal considering the stability of the floating cylinder but allows for using the standard motion capturing system of FloWave. This measurement system further allows for reach the exact same inner water level, so that the draft is equal to the solid case. A difference smaller than 1 mm could be reached. A change of the global $z$-coordinate would also indicate that water flows outside or inside the cylinder. Despite very big rotations, no water exchange could be observed.

In the higher range of the wave frequency, the absorption capacity of the tank is limited and during the test period the reflections can reach the floating object. For the model set-up with the deepest draft filled with water, this has a negative effect on the peak roll motions, but, in all other cases, the influence is relatively small. In general, the chosen frequency band width triggers different responses including all the response frequencies found in the still water experiments.

The soft mooring system is in principal a good choice and with minimal influence on the response of the floating cylinder. The intention was to allow an undisturbed behaviour as long as possible to have a good data set to compare against numerical simulations, which could be reached by focusing on the ramp up time as well as the first oscillations [1]. The downside of the soft approach is that the final position of the model can vary over a wider range in the tank. This can potentially cause different influences due to the reflections as well as expands the needed ramp up time to reach a stable response of the floating structure. In hindsight, the pre-load of all mooring lines should have been increased to minimise the movement in the wave direction. This would further increase the comparability between the different ballast options.

As part of further experiments, the influence of the mooring system will be investigated using the full 360 wave production capability of FloWave. The measurement of the mooring forces would be an option as well as the expansion for irregular wave conditions. Furthermore, the documentation of the behaviour of the inner water body will be improved to allow measuring slamming load depending on the tank geometry [73,74], run-up heights on the walls [59] and detection of the occurrence of breaking waves inside the cylinder.

In the presented investigation, the phase shift between the waves and the response is not the focus of the analysis. In principle, a theoretical wave gauge could be calculated at the position of the floating cylinder and compared with the response. This will be done as part of the further investigations as well as additional wave gauges in a parallel line to the main axis, so that this calculation can be verified.

## 5. Conclusions

The paper presents the results of an investigation of a cylindrical floating object in a wave tank. Regular waves of 0.3 Hz to 1.1 Hz are applied to initiate the response oscillation of the moored floating cylinder. Four different drafts (distance between the bottom point and the still water surface) are tested with two different ballast options, solid and water. Based on the comparison between the liquid and solid, the influence of the sloshing of the stored water can be identified. For all three motions, the differences are relatively small. Due to symmetry, the rotation around the $z$-axis (vertical to the still water surface) is very small. The pitch motion is directly caused by the impacting wave, and large motions over $20°$ are reached. The biggest difference could be identified in the roll motion (rotation around the $x$-axis, which is parallel to the wave direction). These rotational motions are small except for the three deepest drafts for the water ballasted cylinders. In these cases, a change from pitch to roll occurs at a wave frequency in the range of the double response frequency found in the still water test. This is clearly caused be the sloshing of the inside water, and this unstable phenomenon will be further investigated. The full dataset is available based on Gabl et al. [1,2] and can be used as a validation experiment for numerical models to simulate such interactions as well as moving boundary conditions under large motions.

**Author Contributions:** R.G., T.D., E.N., J.S., and D.M.I. are responsible for the conceptualisation of the experimental investigation. R.G., T.D., and E.N. measured the data and analysed the data. R.G. and T.D. wrote the initial draft and E.N., J.S., and D.M.I. reviewed and edited the paper.

**Funding:** This work was supported by the Austrian Science Fund (FWF) under Grant J3918.

**Acknowledgments:** Open Access Funding by the Austrian Science Fund (FWF).

**Conflicts of Interest:** The authors declare no conflict of interest.

## Abbreviations

| | |
|---|---|
| $d$ | cylinder draft (m) |
| $D$ | cylinder diameter (m) |
| $f$ | frequency (Hz) |
| $f_R$ | frequency response (Hz) |
| $f_W$ | frequency wave (Hz) |
| $g$ | gravity acceleration (ms$^{-2}$) |
| $h$ | water depth inside the cylinder (m) |
| $H$ | height of the cylinder (m) |
| $x$ | main wave direction (m) |
| $y$ | orthogonal to the main wave direction (m) |
| FFT | Fast Fourier Transformation |
| QTM | Qualisys Track Manager |
| RMSE | Root Mean Square Error |
| SPH | Smoothed Particle Hydrodynamics |
| VLFS | Very Large Floating Structures |
| WEC | Wave Energy Converter |

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
