# Peer review of "Comparison of a Floating Cylinder with Solid and Water Ballast"

_water, doi:10.3390/w11122487_

Round 1

Reviewer 1 Report

This study presents the comparison of a floating cylinder with solid and water ballast. The influence of the sloshing of the stored water on floating cylinder motion response was identified. They presented some experimental results and discussions with the graphs. The paper was well-written and well-organized. The reviewer recommends publishing the paper after addressing some typological errors and comments.

Further explanations are needed on how the sloshing affects the floating cylinder motion in waves. Whether the sloshing waves were broken or not? Can some test photos be presented in the text for a better explanation this question? The references [19] and [22] are the same. Please delete the reference [22]. There is an extra comma ‘,’ in line 528 in reference [53]. In section 2.2, the experimental photo of the optical motion capture system should be given. In introduction section, the literature review should be more focused on the topic of the paper. Some literatures that were not very relevant to the topic of this paper can be considered for deletion. The response of a floating structure with water and solid ballast options were studied experimentally, respectively, in which the floating structure with water ballast is concerned more widely. In addition to external wave load, the slamming load of the internal fluid on the floating structure should also be of concern. The following two papers are relevant to this issue: “Fluid dynamics analysis of sloshing pressure distribution in storage vessels of different shapes” and “Experimental study on vertical baffles of different configurations in suppressing sloshing pressure”. It is suggested that the load characteristics of internal fluid should also be discussed in future research. Was the force of the mooring line measured? This can also identify the influence of the sloshing of the internal water on floating cylinder motion response.

Author Response

Thank you. We prove a separate document with the point to point response.

Reviewer 2 Report

Review for Manuscript No. water-618821: “Comparison of a Floating Cylinder with Solid and Water Ballast”

The manuscript presents an experimental study of a floating cylinder with solid or water ballast. The manuscript provides a complete literature review of the research background. The experimental methodology is well described. The problem is that it is difficult to understand whether or not (and what kind of) meaningful results/conclusions have been obtained by the presented study.

Some points for the improvement of the manuscript:

Line 4: “a open topped…” -> an open topped…

Line 183: “restricted then” -> restricted than

Line 115: “more then a half” -> more than a half

Line 210: “due” -> due to

The figures are not easy to understand, the figure caption and the many curves. More clear physical explanations for the plots displayed need to be elaborated.

Author Response

(The authors gave the same response as above.)
